# GENERATING ROBUST AUDIO ADVERSARIAL EXAMPLES USING ITERATIVE PROPORTIONAL CLIPPING

## ABSTRACT

Audio adversarial examples, imperceptible to humans, have been constructed to attack automatic speech recognition (ASR) systems. However, the adversarial examples generated by existing approaches usually involve notable noise, especially during the periods of silence and pauses, which may lead to the detection of such attacks. This paper proposes a new approach to generate adversarial audios, which uses Iterative Proportional Clipping (IPC) to significantly limit human-perceptible noise and simultaneously exploits the temporal dependency to resist the defenses based on temporal structure. Specifically, in every iteration of optimization, we use a backpropagation model to learn the raw perturbation on the original audio waveform to construct our clipping. We then impose a constraint on the perturbation at the positions with lower sound intensity across the time domain to eliminate the perceptible noise during the silent periods or pauses. IPC preserves the temporal closeness and trend in the original audio waveform to maintain the temporal dependency. We show that the proposed approach can successfully attack the latest state-of-the-art ASR model Wav2letter+, and only requires a few minutes to generate an audio adversarial example. Experimental results also demonstrate that our approach succeeds in preserving temporal dependency and can bypass temporal dependency based defense mechanisms.

## 1 INTRODUCTION

Due to the recent advancement in machine learning, automatic speech recognition (ASR) systems have been integrated into numerous commercial products. However, researchers have designed adversarial examples to launch targeted attacks towards ASR systems (Vaidya et al. (2015); Carlini et al. (2016)). Under adversarial attacks, ASR systems will recognize the audio inputs as intelligible voice commands, while humans perceive the audio inputs differently. Such attack has proven effective towards ASR systems using Gaussian Mixture Model (GMM) and Hidden Markov Model (HMM) (Lamere et al. (2003)), while recently, ASR systems based on deep neural networks can also be targeted by adversarial examples that slightly perturb the original inputs (Carlini et al. (2016)). Since voice interfaces of ASR products are always listening and have been deployed in sensitive environment, there is an urgent need to study the security and privacy of ASR systems.

While existing research focuses on generating image adversarial examples (Szegedy et al. (2013); Liu et al. (2016); Kos et al. (2018); Arnab et al. (2018); Sharif et al. (2016); Hu & Tan (2017)), these approaches cannot be directly applied to generate effective audio adversarial examples due to the difference in audio and image inputs. In general, we face two major challenges in generating effective audio adversarial examples: **(C1)** Human voices are often interleaved with silent periods due to the pauses in the speech. Inevitably, adversarial examples generated by existing optimization-based approaches (Carlini & Wagner (2018)) could involve non-negligible perturbation on the silent and pause positions, which would affect the quality of adversarial examples and alert the users. **(C2)** When generating adversarial examples against ASR, existing approaches do not consider the inherent voice data property, which allows defense mechanisms (Yang et al. (2018)) that exploit voice data properties to resist such attacks. One important data property that has been used in defense mechanisms is temporal dependency. If the adversarial examples distort temporal dependency of voice data, they can be easily detected.

In this paper, we propose a new approach called Iterative Proportional Clipping (IPC) to generate audio adversarial examples which are not only imperceptible to the humans but also more robust

against existing defense mechanisms. Specifically, we first compute the Connectionist Temporal Classification (CTC) loss (Graves et al. (2006)) of the original audio and the target sentence. Then, we derive the gradient on the original audio through a backpropagation model to be used as the raw perturbation. Moreover, we proportionally clip the perturbation that will be added to the input data. By enforcing a stricter limit on the perturbation at those positions with lower sound intensity in the time domain, our method can introduce negligible noise at silent and pausal positions (addressing **(C1)**). In addition, we limit the perturbed audio waveform proportional to the original audio waveform at all timestamps. By maintaining the trend of the original waveform, our audio adversarial examples succeed to preserve the temporal dependency (TD) in the original audio (addressing **(C2)**). To summarize, we force the perturbation to be within a certain proportion of the original audio waveform at all positions based on iterative proportional clipping, to both restrict the perceptible noise and preserve the TD.

The main contributions of this paper are two fold. First, we launch a successful attack on the latest model of end-to-end ASR system Wav2letter+[1] with a differentiable Mel Frequency Cepstral Coefficient (MFCC) features extraction. Second, we provide a new perspective by considering temporal information in generating robust audio adversarial examples. To the best of our knowledge, we are the first to take TD into account when launching targeted attacks towards ASR systems. Our method generates audio adversarial examples just in a few minutes while prior methods (Carlini & Wagner (2018)) take hours. Compare to Schönherr et al. (2018) which requires the domain-specific knowledge of psychoacoustic hiding, our approach is also more generic and easier to implement in practice.

## 2 RELATED WORK

In their seminal work, Biggio et al. (2013) and Goodfellow et al. (2014) have shown that neural networks are vulnerable to adversarial examples. The followup work (Goodfellow et al. (2014); Fawzi et al. (2016); Liu et al. (2016); Fawzi et al. (2018); Shaham et al. (2018)) further investigates the problem of enhancing the robustness of machine learning models in face of adversarial examples. A substantial amount of efforts have been spent on improving the image recognition models (Kos et al. (2018); Arnab et al. (2018); Sharif et al. (2016); Behzadan & Munir (2017); Huang et al. (2017)), which have seen a significant growth since the inception of deep neural networks (DNN). Meanwhile, recent studies investigate the impact of adversarial examples over text classification (Jia & Liang (2017)) and malware classification (Grosse et al. (2016); Hu & Tan (2017)).

Although we have witnessed the impact of adversarial images on neural networks based systems in recent years, less efforts have been spent on studying the impact of audio adversarial examples towards neural networks based ASR systems. One type of attack approaches is to create an adversarial audio waveform that ASR systems recognize as intelligible voice commands but humans perceive as noise. Vaidya et al. (2015) first explored adversarial examples against ASR systems, which were generated by inputting an audio command into an audio mangler while keeping most of MFCCs intact. The output audio was converted back to a waveform from a lossy inversion of MFCCs, making the output audio unintelligible to a human. Carlini et al. (2016) further extended the black-box approach and constructed white-box attacks via the hidden voice commands on CMU Sphinx speech recognition system (Lamere et al. (2003)), in which they demonstrated HMM-only ASR systems were subject to targeted attack. While these methods can achieve targeted attacks on ASR systems, they cannot generate the adversarial audio waveforms in an end-to-end framework. Zhang et al. (2017) proposed DolphinAttacks, in which they modulated the baseband audio signal onto ultrasound frequency to form inaudible voice commands. This approach requires an expensive attack equipment and highly relies on the software and hardware of the target device. Compared with these approaches, our IPC method is easy to implement with a low cost and can achieve an end-to-end attack framework.

Another type of approaches is to fool neural networks by introducing minor perturbations on the input. Carlini & Wagner (2018) used CTC loss as an objective function and generated adversarial examples using a gradient-descent-based minimization scheme (Carlini & Wagner (2017)). Yuan et al. (2018) successfully embedded transcripts within a popular song. The latest approach considered psychoacoustics to minimize human perception: Schönherr et al. (2018) proposed to gen-

---

[1]https://nvidia.github.io/OpenSeq2Seq/html/speech-recognition/wave2letter.html

erate adversarial examples based on psychoacoustic hiding. They used the hearing thresholds as a guideline to design appropriate manipulations on the input signal by limiting them to be below the thresholds of human perception. Abdullah et al. (2019) utilized domain-specific knowledge of audio signal processing to achieve practical black-box attacks by leveraging the fact that humans interpret discontinuous signals as noisy and hardly discern differences in high frequency signals. These techniques require domain-specific knowledge and complex signal processing. By contrast, optimization-based methods (Opt) (Carlini & Wagner (2018)) are easier to implement, but the generated adversarial examples tend to include widely distributed noise. In this paper, we propose a novel method which extends Opt but limits the total perturbation using iterative proportional clipping, to derive more imperceptible adversarial examples.

## 3 METHOD

In this section, we first introduce our attack workflow for Iterative Proportional Clipping (IPC) attack, and then elaborate on how we use IPC method to generate our adversarial examples. We also explain the imperceptibility of the perturbation on the silent periods. Finally, we analyze the preservation of temporal dependency (TD) of our method.

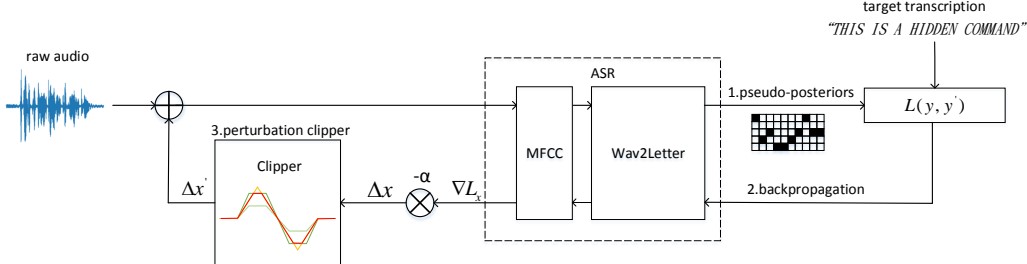

Figure 1: The workflow for generating our adversarial examples.

### 3.1 ATTACK WORKFLOW

The workflow for our adversarial audio generation is shown in Figure 1, which includes three major components: 1) the raw audio is fed into a common ASR system to get the pseudo-posteriors $y$, which is a matrix with each element representing the possibility of each alphabet label at each step. 2) We compute the objective loss $L(y, y')$ in terms of the pseudo-posteriors $y$ and target sentence $y'$ directly, with no need of forced alignment. Then, we compute the gradient $\nabla_x L$ on the input as the raw perturbation through a backpropagation model. 3) The perturbation is then clipped to an allowed range which is derived from the IPC method. Finally, the modified audio added with the clipped perturbation is then fed into the system in the next iteration. Note that the clipping on the gradient in the third component can realize the preservation of TD.

### 3.2 ITERATIVE PROPORTIONAL CLIPPING METHOD

Fast gradient sign method (FGSM) (Goodfellow et al. (2014)) takes the sign of the gradient as the perturbation on the original audio, and the optimization-based method (Opt) (Carlini & Wagner (2018)) uses the gradient itself to perturb the data. Here, we introduce a novel iterative method using the proportionally clipped gradient as the perturbation factor to modify the original audio. Figure 2 illustrates how we construct our clipping in every iteration. Given a natural waveform $x$, we first calculate its Upper Sideline (USL) and Lower Sideline (LSL) as follows:

$$x_{USL} = (1 + \mathbf{B}) \cdot x, \quad x_{LSL} = (1 - \mathbf{B}) \cdot x, \tag{1}$$

where $\mathbf{B}$ is the intensity width of the perturbation in proportion to $x$, the value of which resides between 0 and 1. We use $\mathbf{B}$ to set the maximum thresholds to limit the perturbation. The tolerable perturbation of the original audio will increase with a larger $\mathbf{B}$ value. For each iteration, based on USL and LSL, we update the modified input from the previous iteration by gradient descent to get the new disturbed waveform. Then, we clip the disturbed waveform at those positions where the

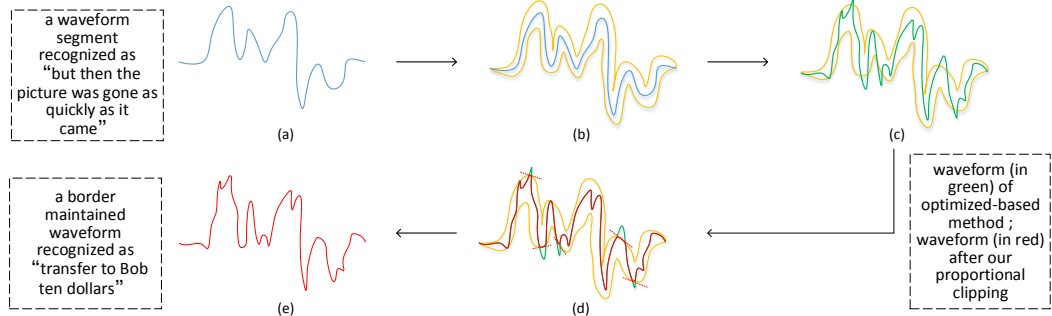

Figure 2: Illustration of the proportional clipping process. Two boundary lines in yellow in (b) which we name Upper Sideline (USL) and Lower Sideline (LSL) respectively, are calculated from the original waveform (a). The line in green in (c) is the disturbed waveform with the perturbation based on Opt. We clip the disturbed waveform in green in (c) at those positions where the sound intensity exceeds the USL or LSL , as shown in (d). The modified waveform in current iteration after our proportional clipping is displayed in red in (e).

sound intensity exceeds the USL or LSL, after which we obtain the modified input for the next iteration. After a number of iterations, we can derive our adversarial examples.

One major advantage of the IPC method is that it limits the perturbation on the silent periods. Since the threshold of each position on the disturbed waveform is proportional to the original waveform, the perturbation on the silent position is constrained and thus more likely to be imperceptible to humans. Existing methods, on the other hand, tend to derive adversarial examples with random noise applied across all positions of the audio.

In addition, our method also preserves the TD of the original waveform through iterative proportional clipping. Existing approaches such as FGSM and Opt-based ones become ineffective against TD based defense mechanisms (Yang et al. (2018)). However, we construct a proportional clipping which aims to preserve the temporal properties including temporal closeness and trend in the original audio. The signal intensity at one timestamp is affected by recent timestamps. Our adversarial audio is more likely to have low signal intensity between two timestamps when both of them have extremely low signal intensities. Besides, the trend in an audio waveform refers to the increase or decrease in signal intensity. Because we limit the intensity variance in a small range at each timestamp, our adversarial audio can preserve the trend in the original audio. To formalize the temporal dependency in our adversarial examples, we compute the cross-correlation coefficient between the original audio sequence and our adversarial example, the result of which exceeds 0.99. Experimental verification on the preservation of the two temporal properties is presented in Section 4. In the end, IPC method provides a mechanism to better control the trend in the output audio waveforms for the purpose of preserving TD.

**Objective loss function.** Similar to Carlini & Wagner (2018), we formulate the problem of constructing an adversarial example as an optimization problem. Given a natural example $x = (x_1, x_2, ..., x_N)$ and any target phrase $y'$, we solve the formulation as follows:

$$\text{minimize } c_1 \cdot \ell(x + \delta, y') + c_2 \cdot |\delta|_2^2$$
$$\text{s.t. } |\frac{\delta_i}{x_i}| < \mathbf{B}, \quad \forall \, 1 \leq i \leq N, \tag{2}$$

where $\ell(\cdot)$ represents the CTC loss. We use the intensity width $\mathbf{B}$ to maintain the proportionality between the perturbation $\delta$ and the original waveform $x$. The parameters $c_1, c_2$ trade off the relative importance of being adversarial and remaining similar to the original audio. Note that we set the restriction on the perturbation in every iteration to produce effective final perturbation results.

## 4 EXPERIMENTAL RESULTS

The presentation flows of the experimental results are summarized as follows. In Section 4.1, we introduce the dataset and adversary model that are used in all of the experiments. In Section 4.2, we introduce our experimental setup on IPC attacks. In Section 4.3, we evaluate the performance of the proposed IPC method from different aspects: first, we perform targeted attacks towards sentences with different lengths to show that IPC method can transform the original audio into arbitrary

target transcription; second, we further provide a visualized comparison between the waveform generated by our approach and other approaches in both time and frequency domains to demonstrate the preservation of the temporal property in the original audio; third, we balance the noise distortion and time cost to develop the optimal attack scheme. In Section 4.4, we describe the experimental setup on TD defense against our adversarial examples. In Section 4.5, we evaluate the preservation of TD in our method, and the results show that our IPC can bypass TD defense effectively.

## 4.1 DATASET AND ADVERSARY MODEL

**Dataset.** LibriSpeech (Panayotov et al. (2015)) is a corpus of approximately 1,000 hours of 16Khz English speech derived from audiobooks from the LibriVox project. It comes with its own training, validation sets, test-clean and test-other sets. We used all available samples to train and validate our ASR system. We generate adversarial examples only using its test-clean set, which contains 2,620 waves with the average duration of 4.294s.

**Adversary model.** Wav2letter (Collobert et al. (2016)) is a simple and efficient end-to-end automatic speech recognition system open sourced by the Facebook AI research team. It combines a standard 1D convolutional neural network, a sequence criterion of typical AutoSegCriterion (ASG) and a simple beam-search decoder. Based on the architecture in Collobert et al. (2016) and Liptchinsky et al. (2017), NVIDIA proposes Wav2letter+[2] which consists of 17 1D-Convolutional Layers and 2 Fully Connected Layers. It extracts log-mel filterbank energies as the input features to the model and uses CTC loss to train the model. In this paper, we implement wav2letter+ in Pytorch as our adversarial model and use beam-search decoder for decoding. Different from the wave2letter+ specification, we use a differentiate MFCC features extraction in front of the ASR model. The output of our model is a sequence of letters corresponding to the speech input. The vocabulary consists of all alphabets (a-z), space, and the apostrophe symbol, a total of 29 symbols including the blank symbol used by the CTC loss.

## 4.2 IPC ATTACK SETUP

We implement the attack using Pytorch. We randomly select one audio file from the test-clean set as the original audio, and the sentence corresponding to another audio file as the target transcription. To make sure the target is arbitrary, we also designate other random sentences as targets. To generate a massive amount of effective adversarial examples quickly and systematically, we conduct attacks in two stages: for the generation of each adversarial example, we first generate a weak adversarial example under the train mode of the ASR system, which is an approximation of the true adversarial example; and then, in the second stage we fine tune it to get the desired adversarial example under the eval mode of the ASR system.

As for parameter settings, we set $c_1 = 1$ and $c_2 = 100$ to represent the case when CTC loss has a larger magnitude. Besides, we set $lr = 1e^{-5}$ in the first stage and $lr = 5e^{-5}$ in the second stage. In order to avoid gradient exploding, we also have a gradient clipping with $grad_{max} = 100$ and $grad_{max} = 10$ respectively in two stages. All experiments are carried out on an Ubuntu Server (16.04 LTS) with an Intel(R) Core(R) i5-6500@ 3.20GHz $\times$ 4, 16G Memory and GTX 1080 GPU.

| Type | Transcribed results |
|---|---|
| Original | but then the picture was gone as quickly as it came |
| Adversarial (short) | open alipay |
| Adversarial (medium) | transfer to bob ten dollars |
| Adversarial (long) | please delete the last transaction record |

Table 1: Examples of the IPC attacks

## 4.3 EVALUATION OF IPC METHOD

Prior works (Vaidya et al. (2015), Carlini et al. (2016), Carlini & Wagner (2018)) have demonstrated that the transfer from a short sentence to a longer one is not a difficult task. The reason is that, differ-

---

[2]https://nvidia.github.io/OpenSeq2Seq/html/speech-recognition/wave2letter.html

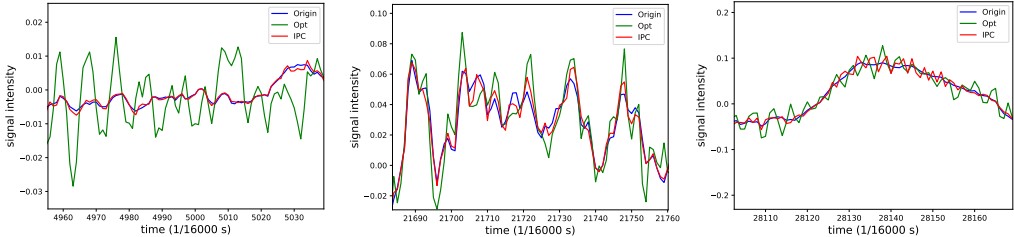

Figure 3: The perturbation at the positions with low sound intensity (left), medium sound intensity (middle) and high sound intensity (right) in time domain.

ent from humans who comprehend each phoneme generally by a dozen frames or so, the machines perceive audios by recognizing each phoneme in at least three HMM states. To show the translated results in different lengths from a general audio waveform, we provide a group of adversarial examples that have been maliciously transcribed in Table 1. We can see different alterations of ASR output toward short, medium, and long utterances, which illustrates that our method can achieve arbitrary targeted attacks from a general audio waveform. More examples are shown in Appendix A.1.

To explore the impact of our IPC method on the waveform segments with different sound intensities compared to Opt, we show the specific perturbation at different positions in a magnified view in Figure 3. For the low sound intensity part in the audio, we can see that our IPC based signal follows the original one closely while the Opt based signal fluctuates significantly. Therefore, the temporal closeness in audio is preserved. Consistent with such phenomenon, our IPC based adversarial examples sound more clear at the silent or the low sound intensity periods. For the middle and high sound intensity parts, IPC and Opt both look similar to the original one, but the former always has smaller perturbation. We infer that our IPC based signal avoids the perturbation of large amplitude at the expense of more frequent variation. Although it makes the audio periods with high intensities sound a little fuzzy, our adversarial example is more acceptable compared to Opt based adversarial examples that contain harsh noise at silent periods. Moreover, the result shows that our signal also maintains the trend of the original audio in the time domain.

We further provide a visualization of the above audio in frequency domain in Figure 4. We first select the signals of 2-5 kHz since the human ear is more sensitive to this band. It shows that Opt produces more noise than IPC, and the Opt signal is more likely to be perceptible to humans. However, our IPC signal is almost the same as the original one. We also visualize the audio in higher frequency bands, where Opt produces obvious noise, which implies that IPC attacks perform better than Opt attacks, and are more robust against defenses which defend against adversarial examples based on high frequency filtering. See Appendix A.2 for more visualizations of the adversarial examples in time and frequency domain.

To investigate the performance with different intensity widths **B** in Eq. (1), we conduct a series of experiments with different intensity widths while generating adversarial examples. We mainly

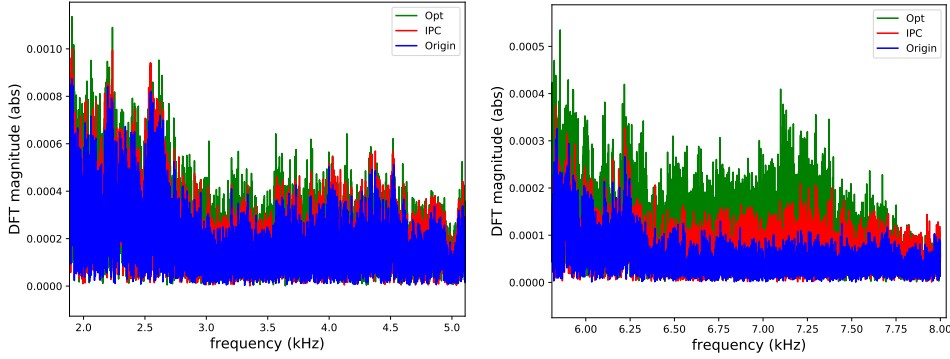

Figure 4: The perturbation at the positions with human sensitive frequency band (left) and high frequency band (right).

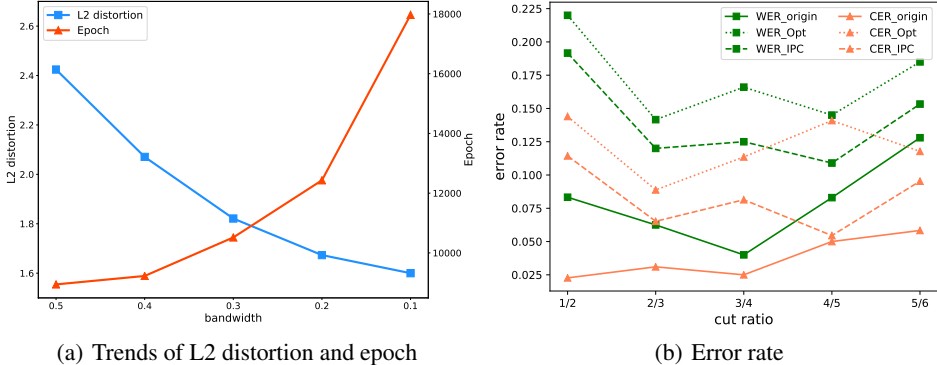

(a) Trends of L2 distortion and epoch     (b) Error rate

Figure 5: (a) Trends of L2 Distortion and epoch as the intensity width gets narrower. (b) Word Error Rate and Character Error Rate of audio examples.

explore the trend of two key metrics with a narrower intensity width, which is shown in Figure 5(a). One metric is the L2 distortion, which is chosen for quantifying the distortion introduced by the perturbation; the other one is the number of epochs required to alter original transcription to the target. We can find that the L2 distortion descends slowly from 2.4 to 1.6 while the number of requisite epochs grows as the intensity width narrows down from 0.5 to 0.1. Specifically, the number of requisite epochs explodes to be more than 5,000 when the intensity width drops to 0.1. To strike a balance between epochs and distortion, we set the intensity width to 0.2 in all the following experiments to generate adversarial examples with a high quality while reducing runtime cost. As the classical paper Carlini & Wagner (2018) has demonstrated, generating a single adversarial example requires approximately one hour of computation time on commodity hardware (a single NVIDIA 1080Ti). In Schönherr et al. (2018), the proposed method requires to compute a matrix which contains the difference with the hearing thresholds at each iteration. They claim it took less than two minutes to calculate the adversarial perturbations with 500 backpropagation steps, but do not specify the exact time for a successful attack. In contrast, to accomplish a successful attack, our time consumption is minute-level, usually in 3-15 minutes on commodity hardware (a single NVIDIA 1080), without any complex computation. We encourage the reader to listen to our adversarial examples to hear how similar they are to the original audio[3].

## 4.4 TD DEFENSE SETUP

Due to the preservation of TD, our adversarial examples are more robust than the state-of-the-art adversarial examples (Carlini & Wagner (2018)). To validate their robustness, we implement attacks using our adversarial examples to target ASR systems equipped with TD defense (Yang et al. (2018)). TD defense gains discriminate power against adversarial examples by utilizing the inconsistency between the prefix of length $l$ of the transcribed results and the transcribed portion of the same length $l$. The audio portion of length $l$ is obtained by cutting the tested audio with different cut ratios $k$. With $k = 0.5$, for example, TD defense will compare the transcription of the first half of audio and the corresponding prefix of the original transcription. Common adversarial examples often involve some unwished losses of temporal information. Our proposed method circumvents the loss of temporal information and therefore can effectively bypass TD defense. We follow the same experimental procedures as Yang et al. (2018), and adopt their evaluation metrics: the area under curve (AUC) of word error rate (WER), AUC of character error rate (CER), and AUC of longest common prefix (LCP).

## 4.5 EVALUATION OF IPC METHOD AGAINST TD DEFENSE

Generally, as TD defense exploits the broken TD in audio data to identify adversarial examples, common Opt based adversarial examples can be detected owing to the inconsistent transcription of the same $k$ portion. To illustrate that the effectiveness of our adversarial examples is not impacted by TD defense, we list some examples of translated results for benign and adversarial audios with the cut ratio $k = 0.5$ in Table 2. We can see that segments of our adversarial examples can be equally

---

[3]https://drive.google.com/open?id=14LSY9x5lEhaVtJGtOKNpeD8tpGHBCSEh

| Type | Transcribed results |
|------|---------------------|
| Original | but then the picture was gone as quickly as it came |
| the first half of Original | but then the picture was to |
| | |
| Adversarial (short) | open alipay |
| First half of Adversarial | open |
| Adversarial (medium) | transfer to bob ten dollars |
| First half of Adversarial | transfer to bob |
| Adversarial (long) | please delete the last transaction record |
| First half of Adversarial | please delete the last |

Table 2: Examples of the TD based detection on IPC method

| $k$ | IPC (Opt) | | |
|-----|-----------|-----|-----|
| | WER | CER | LCP |
| 1/2 | 0.524 (0.930) | 0.507 (0.933) | 0.609 (0.806) |
| 2/3 | 0.770 (0.930) | 0.700 (0.948) | 0.885 (0.826) |
| 3/4 | 0.573 (0.933) | 0.510 (0.938) | 0.835 (0.839) |
| 4/5 | 0.575 (0.955) | 0.553 (0.969) | 0.772 (0.880) |
| 5/6 | 0.755 (0.941) | 0.680 (0.962) | 0.766 (0.858) |

Table 3: AUC scores of different $k$ on IPC and Opt adversarial examples

transcribed with the corresponding transcription of the whole adversarial examples. More results on our adversarial examples against TD defense are shown in Appendix A.4. Moreover, we explore the average WER and CER with different cut ratios under TD defense in Figure 5(b) to verify if the IPC based adversarial examples preserve TD. Compared with Opt, IPC based adversarial examples have lower WER and CER. The error of IPC is caused by the wrongly transcription of the last word. These experimental results demonstrate the IPC method can preserve the temporal information of the original audio. This is because the added perturbation in the audio is nearly proportional to the original audio across different positions, and thus does not change the trend of the audio.

To further investigate the effectiveness on the preservation of TD, Table 3 represents the AUC scores of three basic metrics under different cut ratios on the IPC based adversarial examples. The AUC scores on IPC are mostly distributed between 0.5 and 0.7, which means the classifier with TD defense has poor performance on our adversarial examples. Moreover, TD defense with metrics of WER and CER performs worse than that with LCP. We observe that the AUC scores based on WER and CER of IPC are relatively low and can reach slightly above 0.5. And the AUC score based on LCP reaches above 0.8 under the cut of 2/3 and 3/4 due to the unbalanced proportion of longer and shorter transcribed targets. It is clear that our AUC scores are lower than the scores of Opt method under different cut ratios, which demonstrates the robustness of our method against TD defense due to the preservation of temporal information.

## 5 CONCLUSION

In this paper, we propose a new method based on IPC to achieve targeted attacks on ASR system, which, for the first time, considers the preservation of temporal dependency to generate robust audio adversarial examples. By iteratively performing proportional clipping on the perturbation which we compute from the gradient through a backpropagation model, we force the modified audio waveform to maintain the trend in the original audio. Our experiments show that IPC based adversarial examples can not only fool the state-of-the-art Wav2letter+ model but also bypass the latest temporal dependency based defense. Experimental results also demonstrate that IPC significantly outperforms other voice attack methods, and the added noise on the original audio becomes imperceptible to humans.

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

## A  APPENDIX

### A.1  MORE RESULTS ON IPC ATTACK

Tables 4-6 provide the results of targeted attacks from different original waveforms. We show that they are converted into transcriptions of different lengths for each example.

| Type | Transcribed results |
|---|---|
| Original | but then the picture was gone as quickly as it came |
| Adversarial (short) | what is slang |
| Adversarial (short) | what was that |
| Adversarial (short) | you are acute |
| Adversarial (medium) | i don't anticipate |
| Adversarial (medium) | sunday august sixteenth |
| Adversarial (medium) | why are we to be divided |
| Adversarial (long) | there it clothes itself in word masks in metaphor rags |
| Adversarial (long) | for some time after that i remembered nothing distinctly |
| Adversarial (long) | yes yes she hurried pulling her hand gently away from him |

Table 4: Examples of the IPC attacks from "but then the picture was gone as quickly as it came".

| Type | Transcribed results |
|---|---|
| Original | surely we can submit with good grace |
| Adversarial (short) | i know |
| Adversarial (short) | got it |
| Adversarial (short) | run away |
| Adversarial (medium) | you are positive then |
| Adversarial (medium) | he nods his consen |
| Adversarial (medium) | we suffer stifling pains |
| Adversarial (long) | it will be no disappointment to me |
| Adversarial (long) | tea please matthews butler impassively |
| Adversarial (long) | it will not be safe for you to stay here now |

Table 5: Examples of the IPC attacks from "surely we can submit with good grace".

| Type | Transcribed results |
|---|---|
| Original | there it clothes itself in word masks in metaphor rags |
| Adversarial (short) | i'm happy |
| Adversarial (short) | you did it |
| Adversarial (short) | open the door |
| Adversarial (medium) | there just in front |
| Adversarial (medium) | anders face grew red |
| Adversarial (medium) | now to bed boy |
| Adversarial (long) | tis fine for you to talk old man answered the lean aullen apprentice |
| Adversarial (long) | it will not be safe for you to stay here now |
| Adversarial (long) | run back uncas and bring me the size of the singer's foot |

Table 6: Examples of the IPC attacks from "there it clothes itself in word masks in metaphor rags".

## A.2 MORE VISUALIZATIONS OF THE ADVERSARIAL EXAMPLES IN THE TIME AND FREQUENCY DOMAIN

We show the IPC attack from "but then the picture was gone as quickly as it came" to "why are we to be divided" in the time domain in Figure 6-7, and frequency domain in Figure 8.

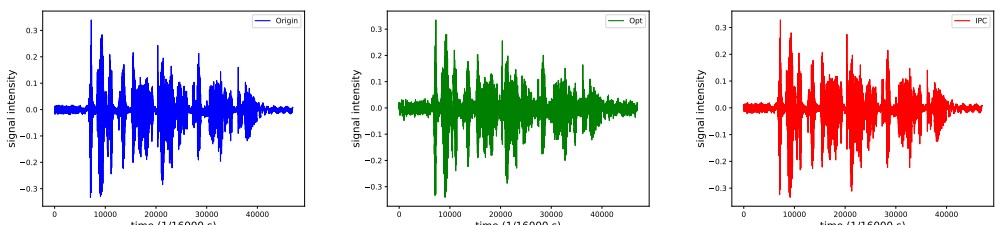

Figure 6: The original audio(left), Opt based adversarial example (middle) and IPC based adversarial example (right) in time domain.

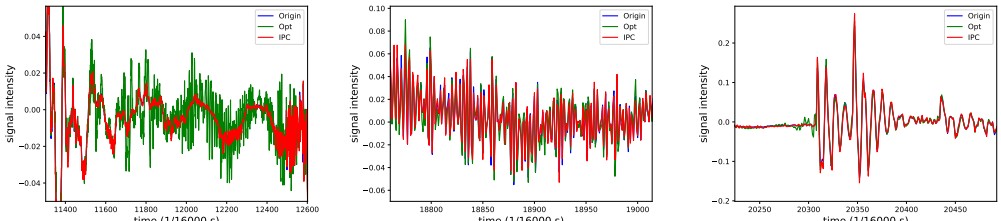

Figure 7: The perturbation at the positions with low sound intensity (left), medium sound intensity (middle) and high sound intensity (right) in time domain.

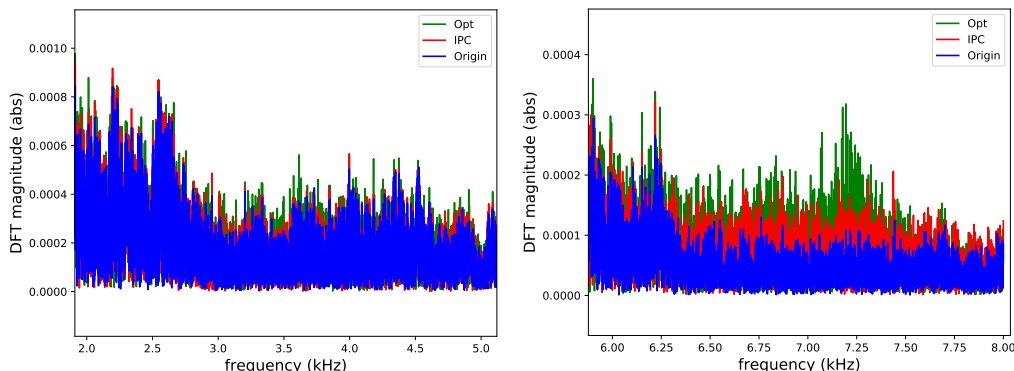

Figure 8: The perturbation at the positions with human sensitive frequency band (left) and high frequency band (right).

We show the IPC attack from "surely we can submit with good grace" to "it will not be safe for you to stay here now" in the time domain in Figure 9-10, and frequency domain in Figure 11.

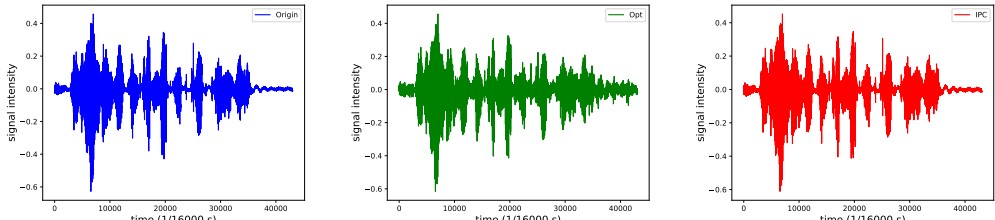

Figure 9: The original audio(left), Opt based adversarial example (middle) and IPC based adversarial example (right) in time domain.

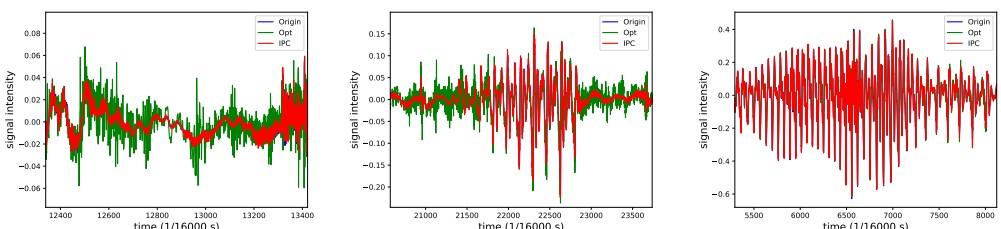

Figure 10: The perturbation at the positions with low sound intensity (left), medium sound intensity (middle) and high sound intensity (right) in time domain.

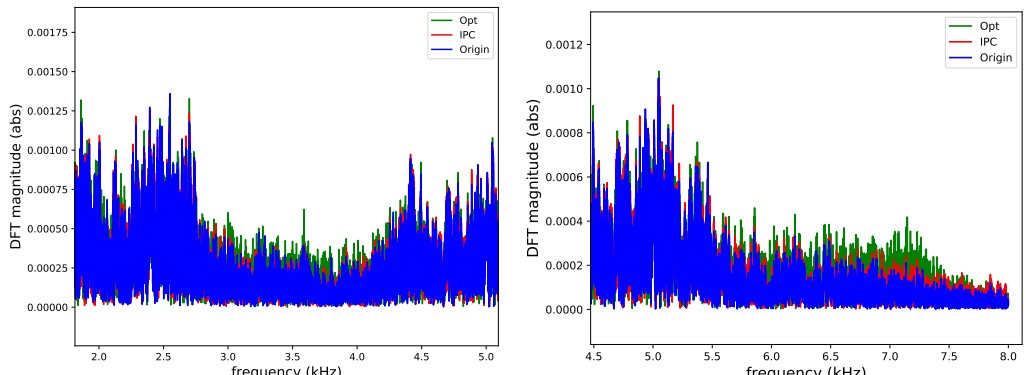

Figure 11: The perturbation at the positions with human sensitive frequency band (left) and high frequency band (right).

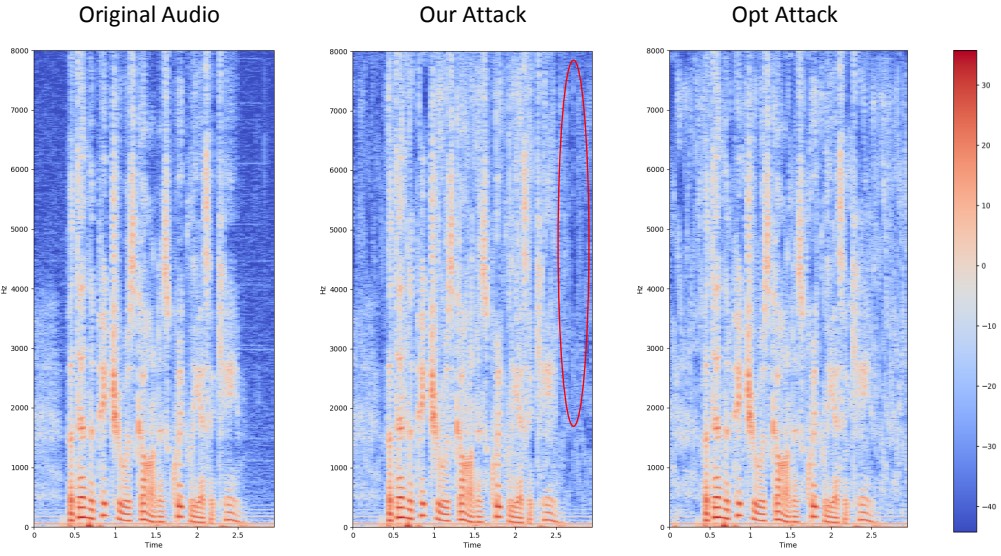

Figure 12: This figure displays an STFT of an audio file, our perturbation of the audio file, Opt's perturbation of the audio file. The red elliptical area shows less noise in silent periods than Opt.

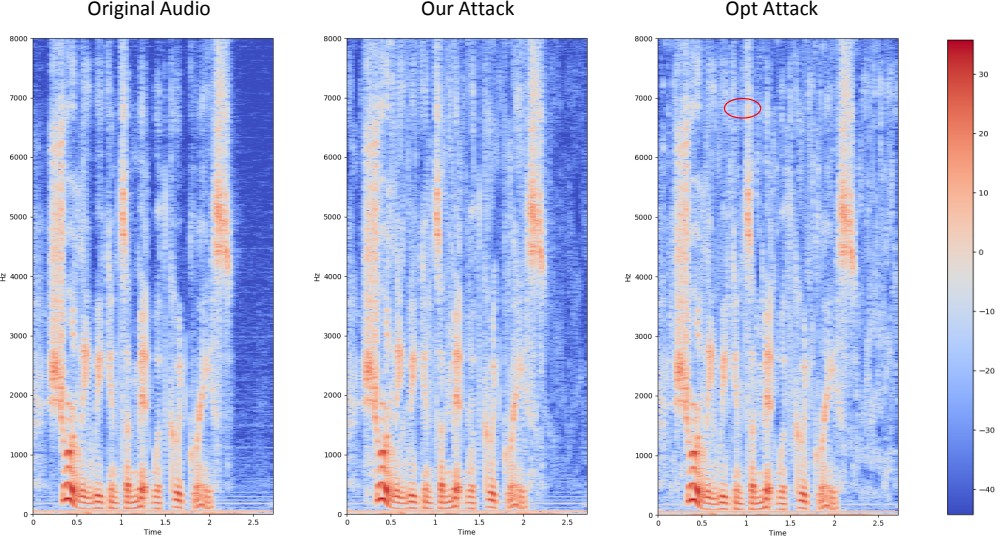

Figure 13: This figure displays an STFT of an audio file, our perturbation of the audio file, Opt's perturbation of the audio file. The red elliptical area shows different frequency components at a certain timestamp.

From spectrograms in Figures 12-13, on one hand, we can see the variance in signal intensity over time (reflected in color) is consistent with that in the original signal in a particular frequency band, which suggests the trend is maintained. On the other hand, there are minimal noise in the silent periods in blue, which makes our adversarial audio perceptually more appealing than Opt based adversarial audio.

### A.3 Results on Opt based adversarial examples against TD defense

To show the Opt method's failure in preserving the temporal dependency, Tables 7-9 provide some additional examples of translated results for benign and Opt based adversarial audios under TD detection. We can see the transcription of the first half of audio always contains different words with the corresponding prefix of the original transcription in all cases, which shows the inconsistency between the prefix of the transcribed results and the transcribed portion of the same length.

| Type | Transcribed results |
|---|---|
| Original | but then the picture was gone as quickly as it came |
| the first half of Original | but then the picture was to |
| | |
| Adversarial (short) | what is slang |
| First half of Adversarial | why this |
| Adversarial (medium) | no my little son she said |
| First half of Adversarial | my little son |
| Adversarial (long) | it will not be safe for you to stay here now |
| First half of Adversarial | it were no bus |

Table 7: Examples of the temporal dependency based detection on Opt method

| Type | Transcribed results |
|---|---|
| Original | surely we can submit with good grace |
| the first half of Original | surely we can submit |
| | |
| Adversarial (short) | i know |
| First half of Adversarial | i ki |
| Adversarial (medium) | why are we to be divided |
| First half of Adversarial | why are we do |
| Adversarial (long) | it will be no disappointment to me |
| First half of Adversarial | it will be conclude |

Table 8: Examples of the temporal dependency based detection on Opt method

| Type | Transcribed results |
|---|---|
| Original | there it clothes itself in word masks in metaphor rags |
| the first half of Original | there it clothes itself in word |
| | |
| Adversarial (short) | i'm happy |
| First half of Adversarial | more |
| Adversarial (medium) | oh i know that's lorne brandon |
| First half of Adversarial | oh i knew that lord |
| Adversarial (long) | a golden fortune and a happy life |
| First half of Adversarial | a golden fortun and a whiping |

Table 9: Examples of the temporal dependency based detection on Opt method

A.4   MORE RESULTS ON IPC BASED ADVERSARIAL EXAMPLES AGAINST TD DEFENSE

Tables 10-12 provide some additional examples of translated results for benign and IPC based adversarial audios under TD detection. We can see the transcription of the first half of audio is almost identical to the corresponding prefix of the original transcription in all cases, which shows the consistency between the prefix of the transcribed results and the transcribed portion with the same length.

| Type | Transcribed results |
| --- | --- |
| Original | but then the picture was gone as quickly as it came |
| the first half of Original | but then the picture was to |
| | |
| Adversarial (short) | what is slang |
| First half of Adversarial | what is |
| Adversarial (medium) | i don't anticipate |
| First half of Adversarial | i don't any |
| Adversarial (long) | there it clothes itself in word masks in metaphor rags |
| First half of Adversarial | there it clothes itself in word |

Table 10: Examples of the temporal dependency based detection on IPC method

| Type | Transcribed results |
| --- | --- |
| Original | surely we can submit with good grace |
| the first half of Original | surely we can submit |
| | |
| Adversarial (short) | i know |
| First half of Adversarial | i know |
| Adversarial (medium) | you are positive then |
| First half of Adversarial | you are pose |
| Adversarial (long) | it will be no disappointment to me |
| First half of Adversarial | it will be no disappoint |

Table 11: Examples of the temporal dependency based detection on IPC method

| Type | Transcribed results |
| --- | --- |
| Original | there it clothes itself in word masks in metaphor rags |
| the first half of Original | there it clothes itself in word |
| | |
| Adversarial (short) | i'm happy |
| First half of Adversarial | i'm happy |
| Adversarial (medium) | there just in front |
| First half of Adversarial | there just |
| Adversarial (long) | tis fine for you to talk old man answered the lean sullen apprentice |
| First half of Adversarial | tis fine for you to talk old man |

Table 12: Examples of the temporal dependency based detection on IPC method

