# OpenReview forum: "Generating Robust Audio Adversarial Examples using Iterative Proportional Clipping"
_ICLR.cc/2020/Conference — Reject_

### Official Review · AnonReviewer3 · 2019-10-20
**Official Blind Review #3**

**Rating:** 3

**Review:**

This paper proposes a technique to generate audio adversarial examples based on iterative proportional clipping. It is essentially an optimization approach via backprop but with a constraint on the magnitude of perturbation in the time domain.  The authors show the effectiveness of the proposed technique to successfully attack the wav2letter+ ASR model and claim to have superior temporal dependency than other existing approaches.  Overall, the work is interesting but I have the following concerns.

1. In terms of analysis, I think it would be more helpful to show results in spectrogram rather than frequency as shown in Fig.4.

2.  The authors claim that the proposed technique is less time-consuming to generate in the abstract. However, there is no discussion and comparison in the experiments.

3.  Since the proposed technique is claimed to be iterative, it would be helpful to demonstrate the "iterative" performance improvement.  How much does the iteration help?  How many iterations does it need to get a good adversarial audio?  How long does it take to accomplish that?  I couldn't find such information in the paper.

4. By listening to the provided audio samples, I would say the quality of the audios are good in general. But in w4, one can actually perceive the distortion in the audio. But overall it is good. Thanks for the samples.

5  The wording and formulation need to get improved.
     --  I found the word "bandwidth" very confusing especially when it comes to the discussion about the time and frequency domain signal processing.  The word "bandwidth" has a well-established meaning in signal processing already.
     --  There is no definition of "TD" throughout the paper -> temporal dependency?
     --  The formulation in Eq.2 is incorrect. $\delta$ in the second term seems to be independent of the first term. shouldn't it be $l(x+\delta, y')$ in the first term?

P.S.  rebuttal read.  I will stay with my score.

**Experience Assessment:**

I have read many papers in this area.

**Review Assessment: Checking Correctness Of Derivations And Theory:**

I assessed the sensibility of the derivations and theory.

**Review Assessment: Checking Correctness Of Experiments:**

I assessed the sensibility of the experiments.

**Review Assessment: Thoroughness In Paper Reading:**

I read the paper at least twice and used my best judgement in assessing the paper.

---

> ### Author Response · Authors · 2019-11-10
> **Response to Review #3**
>
> Thanks for your careful and valuable comments. We address your concerns point by point. We hope to convince you to reassess our paper.
>
> A1: First of all, I think it is helpful to show results in spectrogram. In our paper, we visualize our results in time and frequency domains separately in details. To intuitively show the superiority of our attacks, we select the most sensitive frequency bands of human hearing and compare the results of Opt method and ours, and display them in Fig. 4. But for a more comprehensive understanding of our attacks, we added the comparison in spectrogram and the corresponding analysis in Appendix in our revision (See Fig.12 and Fig.13).
>
> A2: As the classical paper [1] has demonstrated, generating a single adversarial example requires approximately one hour of computation time on commodity hardware (a single NVIDIA 1080Ti), with the same computational power of 1.5-2 NVIDIA 1080 we have used. As mentioned in our contribution, the prior works require an hour-level time cost.
> In [2], the proposed method requires to compute a matrix that contains the difference in dB to the hearing thresholds at each iteration. They claimed it took less than two minutes to calculate the adversarial perturbations with 500 backpropagation steps, but did not specify the exact time for a successful attack. Moreover, for some cases, their methods cannot achieve success even after 5000 iterations. In contrast, to accomplish a successful attack, our time consumption is minute-level, usually in 3-15 minutes, without any complex computation.
> And we add the discussions about time consumptions in prior methods in our revised version.
>
> A3: We implement our attacks by solving the minimization problem. The proportional clipping works as a well-designed noise constraint during the optimization of the objective loss. Therefore, at each iteration, the audio will vary itself across all timestamps towards the direction of a maximized gradient.
>
> In addition, our experiment is carried out with the goal of achieving 0% WER. That means different target texts may require different iterations to achieve the goal. We don’t explore the WER when the goal is not achieved, since these cases indicate a failed attack.
>
> As a result, we need 5000 (stage 1)+500 (stage2 with a beamsearch decoder) iterations to get a five-second audio file in 10 minutes on an Ubuntu Server (16.04 LTS) with an Intel(R) Core(R) i5-6500@ 3.20GHz × 4, 16G Memory and GTX 1080 GPU. For a ten-second audio file, the time consumption, which is within 15 minutes, does not increase significantly.
>
> A5:(1) We replace “bandwidth” with “intensity width” to avoid confusion. Thanks for your suggestions.
> (2)“TD” is the abbreviation form of “temporal dependency”.
> (3)The mistake has been corrected in Eq.2.
>
> [1]Audio Adversarial Examples: Targeted Attacks on Speech-to-Text. Nicholas Carlini, David Wagner. IEEE SPW 2018.
> [2]Adversarial Attacks Against Automatic Speech Recognition Systems via Psychoacoustic Hiding. Lea Schönherr, Katharina Kohls, Steffen Zeiler, Thorsten Holz, and Dorothea Kolossa. NDSS 2019.

---

### Official Review · AnonReviewer2 · 2019-10-23
**Official Blind Review #2**

**Rating:** 3

**Review:**

This paper introduces an adversarial attack mechanism, Iterative Proportional Clipping (IPC), based on a differentiable MFCC feature extractor. In contrast to other audio attacks, the authors claim that the perturbations are well masked and are much less audible.
They also claim to propose the first attack mechanism that takes temporal dependency into account, at least in some intuitive sense as described in Figure 2. Simulations show that state of the art speech recognizers can be attacked using the approach and certain type of defenses, based on temporal structure, can be circumvented. 	The method relies on the idea of using ‘the proportionally clipped gradient’ as a direction estimate for original audio modification. The basic idea is quite sensible and seems to be effective as supported by computational experiments.

The dataset is LibriSpeech and the speech recognizer is based on wav2letter+ -- here the MFCC features are replaced with a differentiable version to allow gradient based adversarial attacks. The experimental section is detailed and the results illustrate the

The fundamental shortcoming with the paper is that while the method is described in detail, there is not much effort in explaining and formalizing what aspects of the proposed method make it perceptually more appealing -- as this seems to be the one of the main motivations of the paper. I would at least expect to see at least some effort to explain the proposed  clipping technique as maintaining the phase properties of the original signal. The authors claim about perceptual properties of the method seem to be somewhat anecdotal and, while I think the method is sensible, just showing the power spectrum is not very informative. Is the proposed method better just because it does good detection of the  silent periods?

I also find the use of technical jargon somewhat ambiguous at occasions (such as the term ‘linear proportionality’)

Minor:

audios -> waveforms  (audio does not have a plural form )

Fig 2 ‘frequency - signal frequency’ labels -- this is probably just an estimate of the power spectrum by DFT magnitude


**Experience Assessment:**

I have published one or two papers in this area.

**Review Assessment: Checking Correctness Of Derivations And Theory:**

N/A

**Review Assessment: Checking Correctness Of Experiments:**

I assessed the sensibility of the experiments.

**Review Assessment: Thoroughness In Paper Reading:**

I read the paper at least twice and used my best judgement in assessing the paper.

---

> ### Author Response · Authors · 2019-11-10
> **Response to Review #2**
>
> A1: Thank you for your comments. We’d like to discuss what aspects of the proposed method make our adversarial audio perceptually more appealing, and why we believe that our method is better than others. We hope we could convince you to reassess our paper.
>
> 1.As mentioned in our paper, our IPC method modifies the signal in time domain, and we preserve the temporal property in the original audio, including temporal closeness, period, trend. This has been explored in another work [1] but in a different domain. These three properties are illustrated as follows:
> First, the signal intensity at one timestamp is affected by recent timestamps. For instance, it’s more likely to have low signal intensity between two timestamps, when both of them present extremely low signal intensities. This has been presented in Fig. 3 in our paper. Second, the period is not taken into consideration when no repeated statement exists in a target text. Lastly, the trend in an audio refers to the increase or decrease in signal intensity. Even with the added noise, our adversarial audio can still preserve the trend in original audio, which can also be seen in Fig. 3. This is because we limit the intensity variance in a small range at each timestamp.
>
> To formalize the temporal dependency in our adversarial examples, we compute the cross-correlation coefficient between the original audio sequence and ours:
>             |   whole     |   low intensity |
>    Opt  |   0.8761    |    0.0230             |
>    Ours|   0.9937    |    0.9987           |
> From the table, the cross correlation coefficient of our IPC method is 0.9937, representing a high degree of relevance while that of Opt is 0.8761, which indicates that our adversarial examples preserve more TD than Opt.
>
> To sum up, our operation preserves temporal dependency ignored by other perceptual based method by maintaining higher consistency with the original audio in time domain. Our experiments have demonstrated that the preservation of temporal dependency can be utilized to bypass corresponding defenses in our paper, and the importance of preserving temporal dependency can be inferred from [2]. And we have added the above explanations about the preservation of temporal properties in our revision.
>
> 2. In terms of perceptual properties, our adversarial examples are perceptually more appealing not only for minimizing the noise of the silent periods. Our method actually improves the entire adversarial audio segment, including these periods with human speech.
>
> Due to the time domain proportional clipping and the fact that amplitude proportional signals have the same frequency component, our adversarial audio causes minimal frequency changes. Due to the minimal frequency changes, our adversarial audio stays closer to the original audio, and thus can be better recognized. We believe the easy recognition of speech tends to cause less awareness of the adversarial noise.
> Intuitively, when we carefully hear our adversarial audio, the periods with high intensities sound a little fuzzy, but this does not cause any recognition issues.
>
> Q2: I also find the use of technical jargon somewhat ambiguous at occasions (such as the term ‘linear proportionality’)
>
> A2: Thank you for pointing out our inappropriate usage of terms. We have replaced some ambiguous technical jargons in our revision.
>
> Q3: Minor
> A3: We corrected the minor mistakes in our revised version.
>
> [1] Deep spatio-temporal residual networks for citywide crowd flows prediction. Junbo Zhang, Yu Zheng, Dekang Qi. AAAI 2017.
> [2] Characterizing Audio Adversarial Examples Using Temporal Dependency. Zhuolin Yang, Bo Li, Pin-Yu Chen, Dawn Song. ICLR 2019.

---

### Official Review · AnonReviewer1 · 2019-10-23
**Official Blind Review #1**

**Rating:** 6

**Review:**

This paper proposed a new method for generating adversarial examples for the task of automatic speech recognition. The authors suggest using their method called Iterative Proportional Clipping, which limits the amount of change we allow at every time-step.

Overall this paper is an incremental research work. The paper is clearly written. The idea is intuitive and well presented.

I have several questions to the authors:
1) Can the authors provide more details regarding the attack setup? Why exactly did you run the attack in two stages? What makes the difference between the train and eval modes? Only on the batch-norm layers?

2) All experiments were conducted in a white-box settings, did the authors try to explore gray/black box settings as well?

3) It seems like there are phase mismatch issues in the generated adversarial examples. Did the authors try to generate adversarial examples using other approximations besides the MFCC to wav approximator? Maybe working at the spectrogram magnitude level?

4) Regarding comment (3). the underlying assumption of adversarial examples is: "it will not be distinguishable from human ears." However, the changes to the signal are highly noticeable. Did the authors try to analyze when is it more or less noticeable, under which settings? Does it depend on the target text?

**Experience Assessment:**

I have published in this field for several years.

**Review Assessment: Checking Correctness Of Derivations And Theory:**

N/A

**Review Assessment: Checking Correctness Of Experiments:**

I carefully checked the experiments.

**Review Assessment: Thoroughness In Paper Reading:**

I read the paper at least twice and used my best judgement in assessing the paper.

---

> ### Author Response · Authors · 2019-11-10
> **Response to Review #1**
>
> Thanks for your comments. Here are our responses.
> A1: About the attack setup, we use a “coarse-to-fine” strategy by executing attacks in two stages to reduce time consumptions.
>
> The wav2letter system performs decoding with a beamsearch decoder based on a 4-gram language model. The modified audio will be iteratively fed into our ASR and be decoded during our generation process to check if it can be recognized as the target text. Note that the iterative decoding process costs much more time than general tensor computation. Therefore, we split our task of generating adversarial examples into two stages by using a “coarse-to-fine” strategy. In stage 1, we decode with a greedy decoder and generate an approximator of the valid adversarial audio. In stage 2, we continuously modify the audio and iteratively decode with a beamsearch decoder. In the end, we get our valid adversarial audio with WER of 0%.
>
> Meanwhile, the differences between the ‘train’ and ‘eval’ modes are on the usage of the batch-norm layers and dropout layers. We set the model in ‘train’ mode in stage 1 because we find it helpful in optimizing our CTCLoss more rapidly without a softmax layer. A normal CTCLoss in ASR train task is minimized in the same manner.
>
> A2: Our attack can be seen as a black box attack for the following reason:
> we use MFCC features through the differentiable feature extraction instead of the log-mel filter bank energy. The feature extraction does not rely on the neural network of ASR systems. Therefore, our attack is effective against other ASR systems as long as every step in the framework is differentiable.
>
> A3: The wav2letter has three types of input features: MFCCs, power-spectrum, and raw wave. In our work, we only utilize the differentiable MFCCs as the input. We believe the other two types of features can also be used. However, we need to avoid the usage of popular python package “Librosa” when processing audio, which blocks the flow of gradient tensor due to the usage of NumPy. We use “torch.rfft” to convert signal to frequency domain in our work.
>
> A4: The variable B in Eq.2 stands for the range of the variations of the original signal. First of all, regardless of the target text, a bigger B indicates more noticeable added noise. That’s why we propose a small B of 0.2. In Fig. 5(a), we analyzed the balance between iterations for attack and distortion according to different B values.
> According to the principle of proportional clipping, regardless of the target text, the noise is mostly concentrated in the places where the signal has a high signal intensity. While the transfer to different target text results in different amounts of noise, whether or not the noise is noticeable mainly depends on the original audio.

---

### Public Comment · ~Yuekai_Zhang1 · 2019-11-01
**Missing trained model  for attack**

In order to reproduce your work faster, is it possible to provid a complete code with model?

---

> ### Author Response · Authors · 2019-11-01
> **Providing the code for attacking our targeted model**
>
> Thanks for your interest in our work . We provide our complete code for our IPC attack in https://github.com/Shmily-coding/ICLR_2020_382 , which also contains our trained Wav2Letter model as a target.

---

### Public Comment · ~Anit_Kumar_Sahu1 · 2019-11-01
**A related work**

Perceptual Based Adversarial Audio Attacks, J. Szurley and Z. Kolter, https://arxiv.org/pdf/1906.06355.pdf. This paper also demonstrates a successful over the air attack.

---

> ### Author Response · Authors · 2019-11-03
> **Thank you for your comment!**
>
> Thank you for your comment! [1] is a reasonable work which is also a perceptual based method, specifically, a psychoacoustic-property-based method. The main idea of [1] is on deriving a weighting factor from the hearing thresholds of the original signal to assign noises to different frequency bands. It's very similar to [2], which utilizes the difference in dB to the calculated hearing thresholds to derive a scaling matrix for gradient in backpropagation.
>
> Although [1] also realizes targeted attack on ASR by generating adversarial examples in time domain, there are significant differences in our paper for the following reasons:
>
> 1) We limit the human perception straightly from the angle of signal intensity in time domain without calculating the hearing thresholds. By iteratively clipping the backpropagated gradient as the perturbation, we directly relate the clipping level to the intensity of the original time-domain signal. Therefore, our approach has the advantages of easy implement and fast generation.
>
> 2) We also consider the adversarial audio's robustness to the latest temporal dependency based defense mechanism, which we believe is important for launching successful attacks.
>
> 3) In addition, our tageted model is the state-of-the-art end-to-end ASR system which is implemented using CNN, instead of the common used RNN-based system.
>
> [1] Perceptual Based Adversarial Audio Attacks. Joseph Szurley, J. Zico Kolter. Arxiv 2019.
> [2] Adversarial attacks against automatic speech recognition systems via psychoacoustic hiding. Lea Schönherr, Katharina Kohls, Steffen Zeiler, Thorsten Holz, and Dorothea Kolossa. NDSS 2019.

---

### Author Response · Authors · 2019-11-11
**Meta-response to all reviewers:**

Thank you all for your time in reviewing our paper. We appreciate the detailed comments and believe all your constructive comments are important for improving this paper. We resubmitted an updated version, which we hope to address your concerns.

In particular, we include the explanation and formalization for temporal dependency, which has not ever been clearly defined in the field of speech recognition. Additionally, we want to emphasize that the purpose of this paper is to introduce a new method operating in time domain to achieve targeted attacks on ASR system. When we are faced with the conflict between frequency operation methods and temporal property based defenses, our paper points to an important direction in investigating different approaches to add perturbations when generating audio adversarial examples.

For the first time, we consider the preservation of temporal dependency to generate robust audio adversarial examples and prove it effective. In contrast to prior methods, our attack is much easier to implement. This can be seen from our attack setup and the pipeline of our approach. Experimentally, we proved the end-to-end and wholely CNN-based Wav2Letter model is vulnerable to such adversarial examples. We also encourage all reviewers (/members of the public on this forum) to check our open source demo. It is neat and runs in just a few minutes on a single GPU. We hope this leads to future collaborations and inspires others in our community.

---

### Decision · Program_Chairs · 2019-12-19

**Decision:**

Reject

**Comment:**

This paper proposes a method called iterative proportional clipping (IPC) for generating adversarial audio examples that are imperceptible to humans. The efficiency of the method is demonstrated by generating adversarial examples to attack the Wav2letter+ model. Overall, the reviewers found the work interesting, but somewhat incremental and analysis of the method and generated samples incomplete, and I’m thus recommending rejection.